# The Association between Diagnosis-to-Ablation Time and the Recurrence of Atrial Fibrillation: A Retrospective Cohort Study

**DOI:** 10.3390/diseases12020038

**Published:** 2024-02-09

**Authors:** Alexandrina Nastasă, Mohamad Hussam Sahloul, Corneliu Iorgulescu, Ștefan Bogdan, Alina Scărlătescu, Steliana Paja, Adelina Pupaza, Raluca Mitran, Viviana Gondos, Radu Gabriel Vătășescu

**Affiliations:** 1Cardiology Department, Elias University Emergency Hospital, 011461 Bucharest, Romania; cardiologie@spitalul-elias.ro (A.N.); stefan.bogdan@umfcd.ro (Ș.B.); 2Faculty of Medicine, “Carol Davila” University of Medicine and Pharmacy, Eroii Sanitari Bvd. 8, 050474 Bucharest, Romania; hussamsahloul21@gmail.com; 3Clinical Emergency Hospital, 014461 Bucharest, Romania; iorgulescu.corneliu@umfcd.ro (C.I.); alina.scarlatescu@gmail.com (A.S.); spitalul@urgentafloreasca.ro (S.P.); miuta-adelina.pupaza@rez.umfcd.ro (A.P.); ralucamitran972@gmail.com (R.M.); 4Department of Medical Electronics and Informatics, Polytechnic University of Bucharest, 060042 Bucharest, Romania; viviana.gondos@hellimed.ro

**Keywords:** atrial fibrillation, catheter ablation, diagnosis-to-ablation time

## Abstract

Background: Catheter ablation (CA) for atrial fibrillation (AF) is superior to antiarrhythmic drugs in maintaining sinus rhythm. Novel evidence suggests that increasing the time between the first diagnosis of AF and ablation, or diagnosis-to-ablation time (DAT), is a predictor for AF recurrence post-ablation. Purpose: Our primary objective was to investigate the relationship between DAT and AF recurrence after a first ablation. Methods: Patients with AF who underwent CA in our center were enrolled consecutively, and a retrospective analysis was performed. DAT was treated as a continuous variable and reported as a median for the group with recurrence and the group without recurrence. DAT was also considered as a categorical variable and patients were stratified into three categories: DAT < 1 year, DAT < 2 years, and DAT < 4 years. Results: The cohort included 107 patients, with a mean age of 54.3 ± 11.7 years. Mean DAT was significantly longer in those with AF recurrence: 4.9(3.06) years versus 3.99(3.5) (*p* = 0.04). The Kaplan–Meier curve revealed a higher likelihood of AF-free status over time for patients with DAT < 2 years compared to those with DAT > 2 years (*p* = 0.04). Cox multivariate analysis indicated that left atrial volume index (LAVI), obstructive sleep apnoea (OSA), and DAT > 2 years were independently associated with AF recurrence after a single AF ablation procedure (*p* = 0.007, *p* = 0.02, and *p* = 0.03, respectively). Conclusion: A shorter duration between the first AF diagnosis and AF ablation is associated with an increased likelihood of procedural success after a single AF ablation procedure.

## 1. Background

Atrial fibrillation (AF) is the most common sustained cardiac arrhythmia [1], and its incidence and prevalence are on the rise globally [2]. The worldwide prevalence of AF is 37,574 million cases (0.5% of the population), and an estimated five million new cases are diagnosed every year [3]. AF is a major public health concern as it imposes a great clinical load, since AF patients have an increased risk for developing stroke, systemic embolism, and heart failure [4,5]. Owing to this, AF places a great financial burden on healthcare systems. The annual costs of AF to European healthcare systems are estimated to range from EUR 660 to EUR 3286 million [6,7,8,9].

Catheter ablation (CA) for atrial fibrillation (AF) is superior to anti-arrhythmic drugs (AAD) in maintaining sinus rhythm [10]. Despite recent technological advancements in AF CA, AF recurrence rates still range from 20% up to 50% [11]. The variation in patient response is due to the heterogeneity of the AF population. Retrospective analyses have identified multiple independent factors associated with AF recurrence, including obstructive sleep apnoea (OSA) [12], metabolic syndrome [13], and hypertension (HTN) [14]. There is novel evidence that increasing the time between first diagnosis of AF and ablation, or diagnosis-to-ablation time (DAT), is also a factor for AF recurrence post-ablation [15,16] and is perhaps one of the most important ones; however, further research is still warranted in order to establish the best moment in the course of AF to perform catheter ablation. The pathophysiology of AF, a progressive disease, sustains the idea that DAT should be considered more carefully and perhaps be included in the guideline indications for CA. Early in its course, AF is an isolated electrical disease initiated by rapidly firing ectopic foci from the pulmonary veins [17]. AF eventually leads to structural remodeling of the atria, characterized by dilation, scarring, and fibrosis [18]. This remodeling results in a more sustained form of AF, contributing to its progression from paroxysmal to persistent AF (PsAF) [19]. It also transforms the condition from an isolated electrical disease into a structural one. Once this progression of AF has occurred, the probability of maintaining sinus rhythm diminishes [20], and once AF becomes persistent, recurrence rates are high post-conversion to sinus rhythm [21]. Not only that, but the progression of AF was reported to be significantly associated with a higher risk of adverse events such as ischemic stroke, systemic embolization, and hospitalization for heart failure [22]. Given this evolutive nature of AF, an early ablation in the course of the disease and, therefore, a shorter DAT, may slow down AF progression from paroxysmal to persistent forms and lead to better outcomes [23,24,25,26]. Our study is aimed at reinforcing the previous results showing DAT is an important predictor of recurrence, and to add some information regarding the optimal time interval to perform the intervention, in other words, how early is early enough, by separating the DAT variable into three categories: DAT < 1 year, DAT < 2 years, and DAT < 4 years.

## 2. Materials and Methods

### 2.1. Study Design and Cohort

A total of 107 patients with paroxysmal or persistent AF who underwent their first CA at our center (Electrophysiology laboratory in the Clinical Emergency Hospital Bucharest) between 2016 and 2019 were enrolled consecutively and a retrospective analysis was performed. The inclusion criteria encompassed current guideline indications. Characteristics including notable comorbidities, cardiovascular risk factors, atrial fibrillation (AF) history, DAT, and prior medication usage, were collected. Additionally, electrocardiograms (ECGs), routine laboratory test results, and echocardiographic data were also collected. Ethical approval was obtained for this study from the Clinical Emergency Hospital Ethics Committee (approval number 29/2016), and all patients signed informed consent. The manuscript adheres to the STROBE guidelines.

### 2.2. Variables

We analyzed the diagnosis-to-ablation time (DAT), defined as the time in years between the initial diagnosis of AF and the first AF CA. In the primary analyses, DAT was treated as a continuous variable in years and reported as a median for both the group with recurrence and the group without recurrence. DAT was also considered as a categorical variable and patients were stratified into 3 categories: DAT < 1 year, DAT < 2 years, and DAT < 4 years.

The primary outcome of interest was AF recurrence after the first AF CA procedure. AF recurrence was defined as the first episode of AF following a 3-month blanking period. Follow-up assessments were performed at intervals of 1, 3, and 6 months initially, which was followed by subsequent evaluations every 6 months, resulting in a median follow-up period of 3 years. The identification of atrial fibrillation (AF) recurrence was accomplished through a combination of clinical interviews and 48-h Holter monitoring. Recurrence was defined as the presence of documented AF, atrial tachycardia, or atrial flutter on electrocardiogram (ECG), or 48-h Holter monitoring, with episodes of the arrhythmia lasting more than 30 s.

Other variables included age at AF diagnosis, sex, type of AF, body mass index (BMI), CHA2DS2-VASc score, and comorbid conditions such as HTN, diabetes, ischemic heart disease (IHD), and OSA. Two echocardiographic parameters were also considered: left ventricular ejection fraction (LVEF) and left atrial volume indexed (LAVI).

### 2.3. Statistical Analysis

The data were analyzed with SPSS v23 (IBM). Analyze-it add-in for Microsoft Excel v23 was also used to aid in the statistical analysis. Continuous variables were presented as the mean with the standard deviation or median (IQR) being reported, while categorical variables were presented as frequencies. The Wilcoxon–Mann–Whitney and Kruskal–Wallis tests were used for continuous variables, while the Fisher test was used for categorical variables for comparison between the group with AF recurrence and the group without recurrence. DAT was considered as a continuous variable in years as well as a categorical variable. A *p*-value of ≤0.05 was considered significant. Kaplan–Meier (KM) survival analysis was carried out to assess time to AF recurrence in those with DAT < 1 year and DAT < 2 years. Variables that have shown significant association with AF recurrence when carrying out the univariate analysis were introduced into a Cox multivariable analysis to ascertain independent predictors of AF recurrence. The criterion for inclusion in the multivariable regression model was a *p*-value ≤ 0.05. Utilizing the Enter Forced Method, Cox proportional regression was employed for the tested variables, incorporating assessments for multicollinearity (tolerance less than 0.1 and a VIF value greater than 10) to validate predictors for atrial fibrillation recurrence. For graphical representation of the performance of DAT as a predictor of arrhythmia recurrence, and in order to assess a good cutoff value for DAT, a receiver operating characteristic (ROC) curve was generated. 

## 3. Results

### 3.1. Cohort Characteristics

The cohort had 107 patients: 57.5% male, with a mean age of 54.3 ± 11.7 years (minimum age 24 yo, maximum 75 yo), and 34% PsAF. 

We included all consecutive patients who underwent first-time radiofrequency catheter ablation (RFCA) with large antral pulmonary vein isolation (PVI) only. The indication for ablation was according to current guidelines: paroxysmal or persistent symptomatic AF and refractory to antiarrhythmic drugs. All patients had a previous CT reconstruction of the left atrium and the pulmonary veins that were integrated into the CARTO 3 system. The patients with previous AF ablation and those with concomitant atrial flutters/atrial tachycardia necessitating additional RA and/or LA ablation besides PVI were excluded. Ablation lesions were guided by contact force (>5 g < 35 g), with a Navistar ThermoCool ST/SF catheter (Navistar, Lisle, IL, USA), 35 watts, 45 degrees, irrigation 17 mL/min, but without an ablation index, as it was the protocol in that period in our center. Following the procedure, the patients did not receive antiarrhythmic medication.

The echocardiographic parameters showed that most patients had an enlarged left atrium with a mean LAVI of 37.2 ± 12.7 mL/m^2^ (moderate dilation category). The LVEF was most often preserved with a mean value of 54.9 ± 8.2%. Structural heart disease was present in our cohort of patients with 7.5% having IHD and 9.4% having tachycardiomyopathy. Heart failure was present in 28.3% of patients. The mean DAT was 4.4 ± 3.3 years with the shortest DAT being 0.5 years and the maximum being 15 years. A total of 20.8% had a DAT of <1 year, 35.8% had a DAT of <2 years and 56.6% had a DAT of <4 years. Time to recurrence was reported in months with a mean of 15.4 ± 14.8 months. The overall recurrence at three years follow-up after the first AF CA was 57.5%. Detailed baseline characteristics are presented in Table 1.

### 3.2. Predictors of AF Recurrence

To establish the association between AF recurrence and other continuous variables, a Mann–Whitney u test was implemented (Table 2). There was a statistically significant difference in the mean LAVI, mean CHA2DS2Vasc, and mean LVEF between the group with AF recurrence and the group without it. Those with AF recurrence had a larger mean LAVI (*p* < 0.0001), a higher mean CHA2DS2 Vasc (*p* = 0.004), and a lower mean LVEF (*p* = 0.009). The association of DAT with AF recurrence was also investigated using the Mann–Whitney u test, and it showed that the mean DAT was significantly longer in those with AF recurrence (*p* = 0.04). The association between categorical variables and AF recurrence was investigated using the Chi square test (Table 3), and the prevalence of HTN, OSA, and PsAF was higher in the group which experienced arrhythmic recurrence.

The impact of DAT on AF recurrence was investigated using the Kruskal–Wallis test, which highlighted a statistically significant difference (*p*-value = 0.04). Those with AF recurrence had a longer mean DAT compared to those without AF recurrence (Figure 1). 

Patients with DAT > 1 year exhibited a higher recurrence rate compared to those with a DAT < 1 year, and the same relationship was observed in patients with DAT > 2 years. Two bar charts were synthesized to better represent these findings (Figure 2). A total of 50% of the AF recurrence group had a DAT > 4 years, while 36% had a DAT > 4 years in the group without recurrence. This finding was not statistically significant. 

The impact of DAT on time to AF recurrence was also analyzed using a Kaplan–Meier survival curve. Given that DAT > 1 year and DAT > 2 years were statistically significant on Fisher’s test while DAT > 4 years was not, DAT > 1 year and DAT > 2 years were the two cutoffs introduced into the KM curve. When examining the KM curve for time to AF recurrence in patients with DAT > 1 year and DAT < 1 year (Figure 3 left panel), we found that the DAT < 1-year group had a cumulative survival without recurrence that appeared higher than that of the DAT > 1-year group (*p* = 0.13). It is important to note that these differences were not statistically significant. On the other hand, when examining the KM curve for time to AF recurrence in patients with DAT > 2 years and DAT < 2 years, we found that the DAT < 2 years group (shorter DAT) had a cumulative survival probability that was higher and, therefore, a higher likelihood of remaining free from AF recurrence over time than that of DAT > 2 years (longer DAT) (*p* = 0.04). Patients with DAT > 2 years had earlier AF recurrence (Figure 3 right panel).

Given that DAT > 2 years rather than DAT > 1 year was the statistically significant cutoff value based on the KM curve, it was introduced into a proportional hazard risk ratio Cox analysis alongside other statistically significant variables in the univariate analysis. The cutoff of the *p*-value used for inclusion in the multivariable regression model was *p* ≤ 0.05 and the Enter Forced Method of the tested variables was used to validate predictors for time to atrial fibrillation recurrence. Left atrial volume indexed (LAVI), obstructive sleep apnea (OSA), and the variable of interest DAT > 2 years were independently associated with AF recurrence after a single AF ablation procedure (*p* = 0.07, *p* = 0.02, and *p* = 0.03, respectively) as seen in Table 4. Diagnosis of the model: CI for exp(B) 95%; probability for variable entry in the model: *p*-value less than 0.05, model chi square 30,363, and model −2 Log Likelihood 487,316. 

ROC analysis for DAT as a continuous variable showed AUC 0.61, *p* = 0.043, 95%CI [0.505;0.723] as shown in Figure 4. A cutoff value of 1.75 offers a sensitivity for predicting recurrence after a single ablation procedure of 87.5% and a specificity of 68%.

## 4. Discussion

Our results show that DAT > 1 year and DAT > 2 years are predictors of AF recurrence. Moreover, patients with DAT > 2 years had earlier AF recurrence, a relationship that we could not demonstrate in patients with DAT > 1 year, an aspect that, combined with the results from the ROC analysis, suggests that a good cutoff value for DAT might be two years. DAT is clearly a modifiable risk factor, but its control requires a nationwide effort to be implemented in order to reduce waiting times, increase access to an electrophysiologist and, therefore, shorten the time from diagnosis of AF to ablation. The recently published randomized clinical trial by Kalman et al. [27] defined ‘early’ AF ablation as occurring within 1 month and ‘delayed’ ablation as taking place within 12 months. This definition of ‘early’ AF ablation is medically quite questionable and definitely not applicable in many centers due to long waiting lists, financial limitations, and difficulty in accessing an electrophysiologist. Also, a factor worth mentioning, and perhaps a confounding factor, could be that DAT is probably also influenced by arrhythmia burden/pattern and symptom severity (patients with initial sporadic episodes and/or who are initially less symptomatic may be referred for CA with a significant delay). Those two circumstances can explain, at least partially, the extended diagnosis-to-ablation time (DAT) in our cohort, which is quite high (mean 4.4 ± 3.3 years).

However, our findings are in line with previous observational studies that showed an increase in DAT is associated with higher AF recurrence rates [28]. In a recent meta-analysis [28], Chew et al., showed that diagnosis-to-ablation times of one year or less were associated with a 27% reduced risk of AF recurrence compared with DAT > 1 year. This was also true in our research. On the other hand, the secondary analysis in the mentioned paper [28] that explored a DAT threshold of three years and also demonstrated lower rates of AF recurrence, was not reproduced in our study, in which the benefits of an earlier intervention seemed to end after a two-year cutoff value. DAT > 4 years was not significantly associated with arrhythmic recurrence in our study. 

Given the temporal nature of atrial remodeling, ablation earlier in the course of the disease allows the treatment of a less remodeled left atrium and provides better procedural outcomes and improves procedural success [23,24,25,26]. Moreover, earlier AF ablation may slow down the progression of AF from paroxysmal to persistent forms [23]. Prior studies have explored the structural modifications associated with the progression of AF. Verma et al., by utilizing voltage mapping to locate areas of left atrial scarring and fibrosis, have demonstrated the relationship between left atrial fibrosis and AF recurrence post-AF CA [29]. The most notable study to also demonstrate this relationship is the prospective multicenter DECAAF (Delayed-Enhancement MRI Determinant of Successful Radiofrequency Catheter Ablation of Atrial Fibrillation). This study has shown a relationship between the degree of atrial fibrosis, as identified by late gadolinium enhancement (LGE), cardiac magnetic resonance imaging (CMR), and AF recurrence rates post-ablation [30]. Chelu et al., also found that a higher level of atrial fibrosis is associated with a three-fold increase in the risk of AF recurrence [31]. DAT can be considered a surrogate marker for atrial remodeling and fibrosis that occurs during the natural evolution of AF. That is to say, a longer DAT might be associated with a more remodeled and fibrotic left atrium and, therefore, lower procedural success and lower likelihood of maintaining normal sinus rhythm. In the present European guidelines, AADs are considered first-line treatment for symptomatic AF, and AF CA is recommended as first-line therapy for a subset of patients such as those with paroxysmal or persistent episodes who are at low risk for procedural complications and recurrence and in patients with tachycardia-induced cardiomyopathy. Only those with tachycardia-induced cardiomyopathy receive a class 1 indication for initial AF CA. Recently, the 2023 American Heart Association (AHA)/American College of Cardiology (ACC)/Heart Rhythm Society (HRS) guidelines have elevated the role of catheter ablation as the initial treatment of AF to 1A, as the role of AF CA is being increasingly recognized in its potential ability to alter the natural history of AF and slow down its progression from paroxysmal to persistent types and, therefore, halting the remodeling process. However, it is important to note that this indication is only for selected patients (generally younger and with fewer comorbidities). The updated European guidelines for AF are set to be published in 2024 and a similar approach in elevating the role of AF CA is to be expected. It is time that the care of electrophysiologists became an early part of the treatment paradigm of AF. It is important to highlight that the detriments of a longer DAT not only include higher recurrence rates, but it also leads to an increased risk of all-cause mortality and heart failure hospitalization as was demonstrated by Bunch et al. [32]. The EAST-AFNET (Early Treatment of Atrial Fibrillation for Stroke Prevention) 4 trial which defined early AF as AF being diagnosed within one year, has shown that an early rhythm control strategy, which also included AF CA, was associated with a lower risk of death from cardiovascular causes, stroke, or hospitalization for heart failure or acute coronary syndrome than usual care (rate control without rhythm control) over a follow-up period of more than five years [33]. On the other hand, another study by Kawaji et al., found that there is no significant difference in five-year clinical outcomes between the short (≤3 years) and long (>3 years) DAT groups, which included ischemic stroke, HF hospitalization, and death [34]. With these conflicting results, more research is required to explore the impact of DAT on hard endpoints such as heart failure and mortality.

Our research reinforces the idea that a shorter DAT and intervention earlier in the course of the disease is beneficial for patients and is associated with lower recurrence rates. It is important to mention that the relatively high recurrence rate found in our study is similar to those reported in that period of time after a single procedure of PVAI [35], given the fact that the RF lesions were less effective then, in the era before strategies for a more homogenous lesion set for PVI by RFCA were deployed such as CLOSE protocol [36] or high-power/short duration [37]. New advancements in ablative technologies emerged after 2019, thus significantly improving the success rates of the ablative procedure.

## 5. Limitations

Several limitations are to be reported. Firstly, this was an observational retrospective study aiming to identify the impact of DAT on AF recurrence. Secondly, the number of patients included in the study was limited, only 107 patients, because in an attempt to have a homogeneous group, we excluded patients with re-interventions, additional ablation lines, as well as those with PVI by different thermal energy sources like cryoablation. Thirdly, the patients included in the final cohort had undergone ablation between 2016 and 2019, so the recurrence rate after the first procedure is quite high, but in line with data from registries published during that time interval [36], especially if we count our prolonged DAT.

## 6. Conclusions

A shorter duration between the first AF diagnosis and AF ablation is associated with an increased likelihood of procedural success after a single AF ablation procedure. Further research is warranted in order to establish the best window of opportunity for catheter ablation in the course of the disease.

## Figures and Tables

**Figure 1 diseases-12-00038-f001:**
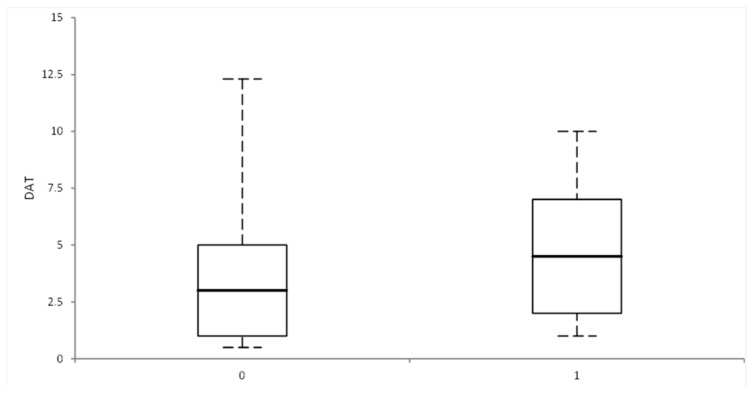
Graphical representation of the difference between median DAT in those with AF recurrence and those without, evaluated using the Kruskal–Wallis test; X axis: 0 = no recurrence, 1 = with recurrence; Y axis = diagnosis-to-ablation time (in months).

**Figure 2 diseases-12-00038-f002:**
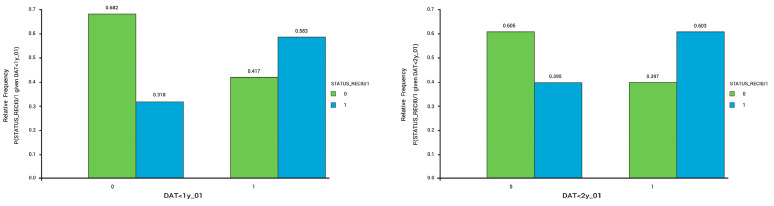
Bar chart comparing recurrence status between those with DAT < 1 year and those with DAT > 1 year (left panel, *p* = 0.03) and between those with DAT < 2 years and those with DAT > 2 years (right panel, *p* = 0.04); Green = no recurrence; Blue = with recurrence.

**Figure 3 diseases-12-00038-f003:**
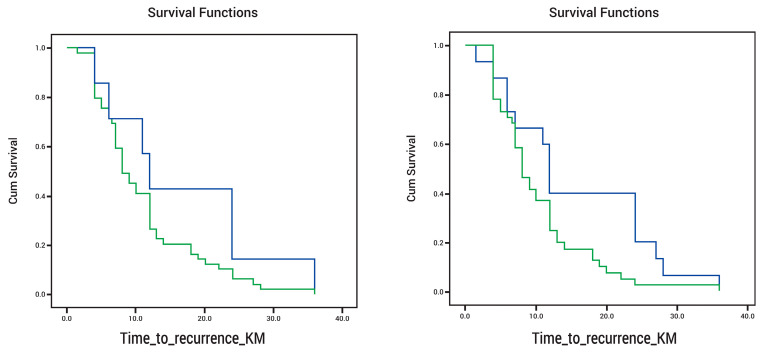
(**Left panel**) Kaplan–Meier survival curve for time to AF recurrence in patients with DAT > 1 year (green) and DAT < 1 year (blue), *p* = 0.13; (**Right panel**) Kaplan–Meier survival curve for time to AF recurrence in patients with DAT > 2 years (green) and DAT < 2 years (blue), *p* = 0.04.

**Figure 4 diseases-12-00038-f004:**
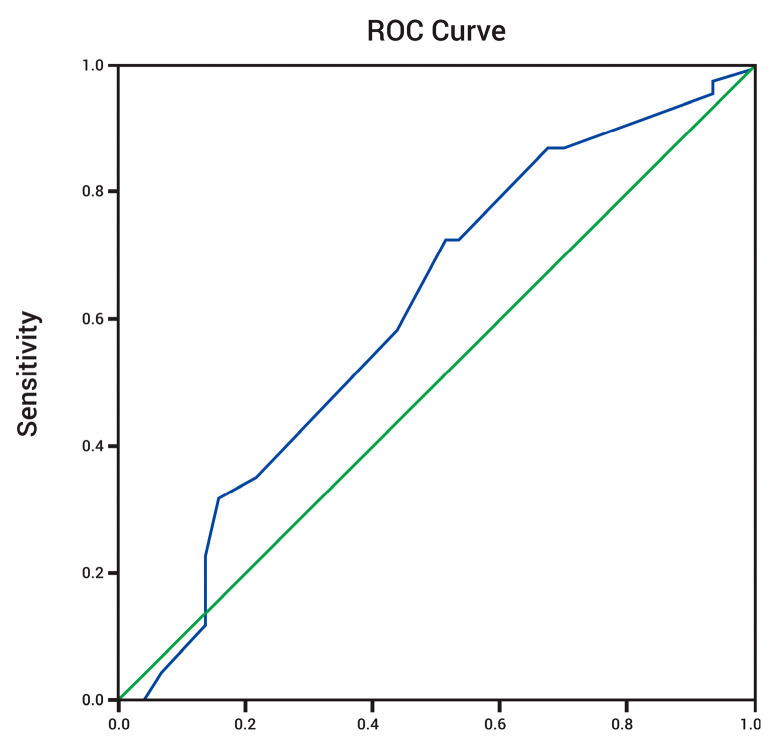
ROC curve for DAT AUC 0.61, *p* = 0.043, 95%CI [0.505;0.723].

**Table 1 diseases-12-00038-t001:** Baseline characteristics of the studied lot.

Patient Characteristics	Mean (SD)/%	Mean (SD)/%	Mean (SD)/%	*p*
	All	DAT < 2 Years	DAT > 2 Years	
Age (years)	54.3 (11.7)	49.1 (12.4)	57.2 (10.3)	0.01
BMI	28.5 (4.5)	29 (4.3)	28.5 (4.6)	0.9
LAVI (mL/m^2^)	37.2 (12.7)	34.3 (8.9)	38.8 (14.2)	0.08
LAV (mL)	75.8 (26.7)	70.34 (18.4)	78.8 (30.1)	0.11
LVEF (%)	54.9 (8.2)	57.1 (5.6)	53.7 (9.1)	0.04
DAT (years)	4.4 (3.3)	1.35 (0.5)	6.2 (2.9)	0.00
Time to recurrence (months)	15.4 (14.8)	15.6 (10.5)	10.2 (6.7)	0.03
Male	57.5%	57%	57.3%	1.00
HTN	47.2%	39%	51%	0.31
IHD	7.5%	2.6%	10.2%	0.25
HF	28.3%	26%	29%	0.82
Dyslipidemia	56.6%	47%	61%	0.16
Type II Diabetes	12.3%	15.7%	10.2%	0.53
OSA	33%	37%	31%	0.66
Smoking	17%	18.4%	16.1%	0.79
Tachycardiomyopathy	7.4%	18%	26.4%	0.47
PsAF	34%	31.5%	35.2%	0.83

**Table 2 diseases-12-00038-t002:** Continuous variables and association with AF recurrence evaluated (WMW).

Continuous Variables	No Recurrence Group	Recurrence Group	*p*-Value
Age (years)	53.1 (11.7)	55.5 (11.8)	0.27
BMI	28.13 (443)	28.9 (4.7)	0.29
CHA2DS2Vasc	1.1 (1.3)	1.7 (1.3)	0.004
Left atrial volume indexed	31.9 (7.72)	42.05 (14.40)	<0.0001
LVEF	56.8 (6.4)	53.3 (9.3)	0.009
DAT (years)	3.99 (3.5)	4.9 (3.06)	0.04

**Table 3 diseases-12-00038-t003:** Categorical variables and association with AF recurrence (univariate analysis).

Categorical Variables	Odds Ratio	CI	*p*-Value
Male sex	0.96	[0.4;2.0]	0.5
Hypertension	2.7	[1.2;6.1]	0.009
Ischemic heart disease	0.88	[0.2;3.7]	0.57
Heart failure	1.8	[0.7;3.6]	0.12
Type 2 diabetes	3.4	[0.8;13.1]	0.057
Obstructive sleep apnea	3.2	[1.3;7.7]	0.006
Persistent AF	2.85	[1.2;6.7]	0.012
DAT > 1	3	[1.1;8.1]	0.024
DAT > 2	2.32	[1.0;5.2]	0.03
DAT > 4	1.7	[0.8;3.8]	0.1

**Table 4 diseases-12-00038-t004:** Independent predictors of arrhythmia recurrence according to Cox multivariable regression.

Variables	B	SE	Wald	df	Sig.	EXP (B)	95.0% CI for EXP (B)	
							Lower	Upper
Left atrium volume indexed	0.028	0.010	7.253	1	0.007	1.028	1.008	1.049
EF	−0.001	0.017	0.006	1	0.938	0.999	0.966	1.033
AF type	−0.444	0.287	2.392	1	0.122	0.641	0.365	1.126
DAT_2y	−0.681	0.315	4.677	1	0.031	0.506	0.273	0.936
Obstructive sleep apnea	−0.645	0.280	5.287	1	0.021	0.525	0.303	0.909

## Data Availability

The data presented in this study are available in this article.

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
