# Peer review of "The Association between Diagnosis-to-Ablation Time and the Recurrence of Atrial Fibrillation: A Retrospective Cohort Study"

_diseases, 2024, doi:10.3390/diseases12020038_

Round 1
Reviewer 1 Report
Comments and Suggestions for Authors
Atrial fibrilation (AF) is a cardiovascular disorder affeting 0.5 of the world population, with an estimated five millions new cases a year. population with AF have five fold risk of ischemic stroke a three-risk increased risk for heart failure and 40% t0 90% increased mortality than that withought. Catheter ablation (CA) fo AF is superior to antiarrthythmic drugs in mantaining sinus rhythm. Novel evidence suggest that increasig the time between the first diagnosis of AF to ablation , or disgnosis to ablation time (DAT) is a predidct for recurrence pst ablation. The primary objectice was to investigate the relationship betwen DAT an AF recurrence after a first ablation. In 107 patients includedf mean DAT was significantly longer in those with AF recurrence. Kaplan-Meyer are revealed AF-free sate over time for patients for DAT<2 years compared to those DAT>2 years (P=0,04) In conclusion a shorter duration between the first AF diagnosis and ablation is associated wiht an increae likehood of procedure success after a single AF procedure and 2 years might be a good cutoff value. The paper is interested althoug the sample is shorter. Others longer studies should be done to confirm those findings, specialy evaluating hard end-points . I suggest some few modifications: Suprress the word Introduction to Background; Don´t put references in the abstract.; In concluson to the end of article meke it more direct, as the same as in abstract. The followe text, if important , should be included in the discussion
Suppress Introduction n the abstract. Suppress reference in abstract. LAVI and OSA are abbreviation and should put full name of the diseases. In Conclusion, after t he end of article include the test as the same in abstract. If The rest of te statement are important to advise your population it may include in the discussion.
Author Response
Dear Reviewer 1,
Thank you so much for taking the time to analyse our work and for your valuable comments, we tried as best as possible to answer them point by point below:
- We replaced the word “Introduction” with “Background” and removed the references from the abstract;
- We made the conclusion more concise as suggested
- We put LAVI (left atrial volume indexed) and OSA (obstructive sleep apnea) in full name
Reviewer 2 Report
Comments and Suggestions for Authors We suggest that the study group perform a subpopulation analysis (patients with cardiomyopathy, ischemic heart disease) including echocardiographic analysis (assessment of left atrial diameter and diastole) to indicate a possible biomarker of recurrence of atrial fibrillation Comments on the Quality of English LanguageGood quality
Author Response
Dear Reviewer 2,
Thank you so much for taking the time to analyse our work and for your valuable comments, we tried as best as possible to answer below:
We suggest that the study group perform a subpopulation analysis (patients with cardiomyopathy, ischemic heart disease) including echocardiographic analysis (assessment of left atrial diameter and diastole) to indicate a possible biomarker of recurrence of atrial fibrillation
- The frequency of ischemic heart disease (IHD) was quite low in our cohort, only 7.5% of the patients had this comorbidty (8 patients), that is why a subgroup analysis including only ischemic cardiomyopathy was not performed; we tested to see if the recurrence rate was different in the patients with and without IHD, but it was not, as shown in Table 3 (Fisher’s exact test showed the p value = 1,00)
- The only echocardiographic parameters available for this lot were: LA volume, and LA volume indexed (LAVI) and LV ejection fraction (LVEF); Cox multivariable regression showed that LAVI was an independent predictor of AF recurrence after ablation (p=0.007) as shown in Table 4.
- If we included any type of structural heart disease/cardiomyopathy vs structurally normal heart, there was still not a statistically significant difference probably due to the small number of patients (p=0.25)
Reviewer 3 Report
Comments and Suggestions for Authors
Dear Editor,
Thank you for the opportunity to review the manuscript entitled “A diagnosis-to-ablation time less than 2 years increases success rates after catheter ablation for atrial fibrillation” worth being published in Diseases following major revision.
The authors present an interesting retrospective analysis on the association between association between diagnosis-to-ablation time and the recurrence of atrial fibrillation. The authors found that there was a higher mean diagnosis-to-ablation time among patients with recurrence of atrial fibrillation. Based on their findings, the authors suggest that a shorter duration between the first diagnosis of atrial fibrillation and subsequent ablation is associated with an increased likelihood of procedural success rate. I recommend the acceptance of this review for publication following a major revision. Predominantly, the authors need to improve their statistical analysis and reporting of methods and results. In addition, there is no information on whether ethical approval has been obtained for this study and the manuscript does not adhere to the STROBE guidelines. Please find my comments in the order as they appear in the manuscript below.
Title
#1 The title should clearly state that this study is a retrospective study and should be more neutral. For example: “The association between diagnosis-to-ablation time and the recurrence of atrial fibrillation: a retrospective cohort study.”
Abstract
#2 Please remove the references from the abstract.
#3 Please state the healthcare setting. It is not clear where the study was conducted.
#4 The analysis performed is not entirely clear. The authors should conduct a logistic regression analysis with the diagnosis-to-ablation time as the independent variable, the recurrence of atrial fibrillation as the dependent variable and several potential confounding variables. Only presenting P-values is not sufficient.
#5 The suggestion of the cutoff value is a bit strong for the conclusion of an abstract. I suggest the authors remove it.
Introduction
#6 The introduction should be re-structured. I suggest that the authors first provide an overview about the “problem” behind atrial fibrillation. For example, what is the incidence? What are the risks for individual patients and what is the health-economic burden of atrial fibrillation?
#7 Currently, the introduction has a strong notion of a discussion. This needs to be revised. The authors clearly need to state the current gap in literature regarding the association between diagnosis-to-ablation time and the recurrence of atrial fibrillation and clearly need to state their study’s objective and hypothesis.
Materials and Methods
#8 Please list the ethical review board name and approval number.
#9 Please state if the manuscript adheres to the STROBE guidelines.
#10 The inclusion criteria need to be described in more detail. It is currently also unclear whether patient’s aged <18 were included.
#11 “The primary variable of interest” sounds like a dependent variable. However, I assume the authors used DAT as their independent variable. This needs to be revised.
#12 “Other variables included ..” – It is unclear whether the listed variables were used as covariates in the confounding model. I suggest the authors perform a multivariable logistic regression analysis as the primary analysis and include the covariates listed.
#13 The information in paragraph 2.4 seems a bit out of place. I suggest this information should be included in the study population.
Results
#14 The baseline characteristics displayed in Table 1 should be presented in the full study cohort (as it is done by the authors) and by the exposure groups.
#15 As already stated, the primary analysis needs to be revised. The authors are requested to perform a multivariable analysis on the association between diagnosis-to-ablation time and the recurrence of atrial fibrillation. Instead of Wilcoxon-Mann-Whitney and Kruskal-Wallis or Fisher tests, I prefer that the authors perform regression analyses and display the odds ratios.
#16 Please also revise Tables 2 and 3, respectively, both of which should show odds ratios.
#17 For the categories of the diagnosis-to-ablation time, the authors are asked to provide odds ratios in the text.
Discussion
#18 I suggest that the authors restructure the discussion towards a more results-focused discussion. For example, at the beginning of the discussion, the authors should focus on stating their main findings.
#19 It is not entirely clear how the authors explain the results of their categorical DAT variable.
Limitations
#20 The limitations section should be expanded: What about data availability, generalizability, heterogeneity of study groups, residual confounding, and other potential biases? Can the authors please comment on this?
Conclusions
#21 Based on the revised analysis, I suggest that the authors change the conclusion statement.
Comments on the Quality of English LanguageQuality of English language is sufficient.
Author Response
Dear Reviewer 3,
Thank you so much for taking the time to analyse our work and for your valuable comments, we tried as best as possible to answer them point by point below:
Thank you for the opportunity to review the manuscript entitled “A diagnosis-to-ablation time less than 2 years increases success rates after catheter ablation for atrial fibrillation” worth being published in Diseases following major revision.The authors present an interesting retrospective analysis on the association between association between diagnosis-to-ablation time and the recurrence of atrial fibrillation. The authors found that there was a higher mean diagnosis-to-ablation time among patients with recurrence of atrial fibrillation. Based on their findings, the authors suggest that a shorter duration between the first diagnosis of atrial fibrillation and subsequent ablation is associated with an increased likelihood of procedural success rate. I recommend the acceptance of this review for publication following a major revision. Predominantly, the authors need to improve their statistical analysis and reporting of methods and results. In addition, there is no information on whether ethical approval has been obtained for this study and the manuscript does not adhere to the STROBE guidelines. Please find my comments in the order as they appear in the manuscript below.
- Ethical approval has been obtained for this study from the Clinical Emergency Hospital Ethics Committee and all patients signed an informed consent.
- We checked the Strobe guidelines and modified the manuscript accordingly
Title
#1 The title should clearly state that this study is a retrospective study and should be more neutral. For example: “The association between diagnosis-to-ablation time and the recurrence of atrial fibrillation: a retrospective cohort study.”
- We changed the title as you suggested, we also believe it is better this way;
Abstract
#2 Please remove the references from the abstract.
- We removed the references from the abstract
#3 Please state the healthcare setting. It is not clear where the study was conducted.
- We added the specifics of the EP center where the analysis was performed : Electrophysiology laboratory in the Clinical Emergency Hospital Bucharest (it is one of the largest volume centers in our country, one of the few that performs complex ablations, including VT, epicardial approach procedures, and one of the two centers in eastern europe that has a magnetic navigation system (Stereotaxis). Two of the authors (myself, Nastasa Alexandrina and dr Bodgan Stefan are currently working in another center, in Elias hospital but in that period of time 2016-2019 we where both still working in the Emergency hospital, so the data is from a single center.
#4 The analysis performed is not entirely clear. The authors should conduct a logistic regression analysis with the diagnosis-to-ablation time as the independent variable, the recurrence of atrial fibrillation as the dependent variable and several potential confounding variables. Only presenting P-values is not sufficient.
- Cox proportional regression using Enter Forced Method of the tested variables (including multicollinearity testing (tolerance less than 0,1 and VIF value greater than 10)) was used to validate predictors for time to atrial fibrillation recurrence. The cut-off of the p value used for inclusion in the multivariable regression model was p ≤0.05. Diagnosis of the model: CI for exp(B) 95%, probability for variable entry in the model: p value less than 0.05, model chi square 30,363, model −2 Log Likelihood 487,316. Model for prediction of time to AF recurrence in 107 consecutive patients is shown in the Table 4:
- Table 4. Independent predictors of arrhythmia recurrence according to Cox multivariable regression
Variables |
B |
SE |
Wald |
df |
Sig. |
EXP (B) |
95.0% CI for EXP (B) |
|
Lower |
Upper |
|||||||
Left atrium volume indexed |
,028 |
,010 |
7,253 |
1 |
,007 |
1,028 |
1.008 |
1.049 |
EF |
-,001 |
,017 |
,006 |
1 |
,938 |
,999 |
,966 |
1,033 |
AF type |
-,444 |
,287 |
2,392 |
1 |
,122 |
,641 |
,365 |
1,126 |
DATlt2y_01 |
-,681 |
,315 |
4,677 |
1 |
,031 |
,506 |
,273 |
,936 |
Obstructive sleep apnea |
-,645 |
,280 |
5,287 |
1 |
,021 |
,525 |
,303 |
,909 |
#5 The suggestion of the cutoff value is a bit strong for the conclusion of an abstract. I suggest the authors remove it.
- We removed the cutoff value, as you suggested
Introduction
#6 The introduction should be re-structured. I suggest that the authors first provide an overview about the “problem” behind atrial fibrillation. For example, what is the incidence? What are the risks for individual patients and what is the health-economic burden of atrial fibrillation?
- We modified the introduction accordingly
#7 Currently, the introduction has a strong notion of a discussion. This needs to be revised. The authors clearly need to state the current gap in literature regarding the association between diagnosis-to-ablation time and the recurrence of atrial fibrillation and clearly need to state their study’s objective and hypothesis.
- We modifed the introduction as you suggested:
Atrial fibrillation (AF) is the most common sustained cardiac arrhythmia [1] and its incidence and prevalence are on the rise globally [2]. The worldwide prevalence of AF is 37,574 million cases (0.5% of the population) and an estimated 5 million new cases are diagnosed every year [3]. AF is a major public health concern as it imposes a great clinical load since AF patients have an increased risk for developing stroke, systemic embolism and heart failure [4,5]. Owing to this, AF places a great financial burden on healthcare systems. The annual costs of AF to European healthcare systems is estimated to range from €660–€3,286 million [6], [7], [8], [9]. There is novel evidence that increasing the time between first diagnosis of AF and ablation, or diagnosis-to-ablation time (DAT) is also a factor for AF recurrence post ablation [15],[16] and perhaps one of the most important ones, but further research is still warranted in order to establish the best moment in the course of AF to perform catheter ablation. [...] Given this evolutive nature of AF, an early ablation in the course of the disease and therefore a shorter DAT may slow down AF progression from paroxysmal to persistent forms and lead to better outcomes [23], [24], [25], [26]. Our study is aimed to reinforce the previous results showing DAT is an important predictor of recurrence, and to add some information regarding the optimal time interval to perform the intervention, in other words, how early is early enough, by separating the DAT variable into 3 categories : DAT<1 year, DAT<2 years and DAT<4 years.
Materials and Methods
#8 Please list the ethical review board name and approval number.
- Ethical approval has been obtained for this study from the Clinical Emergency Hospital Ethics Committee (approval number 29/2016) and all patients signed an informed consent.
#9 Please state if the manuscript adheres to the STROBE guidelines.
- the manuscript adheres to the STROBE guidelines
#10 The inclusion criteria need to be described in more detail. It is currently also unclear whether patient’s aged <18 were included.
- We included all consecutive patients who underwent first-time radiofrequency catheter ablation (RFCA) with large antral pulmonary vein isolation (PVI) only. The indication for ablation was according to current guidelines, paroxysmal of persistent symptomatic AF, refractory to antiarrhythmic drugs. All patients had a previous CT reconstruction of the left atrium and the pulmonary veins that were integrated into the CARTO 3 system. The patients with previous AF ablation and those with concomitant atrial flutters / atrial tachycardia necessitating additional RA and/or LA ablation besides PVI were excluded. Following the procedure, the patients did not receive antiarrhythmic medication.
- Minum age was 24 yo, mean 54,34 years and maximum 75 years.
Descriptive Statistics |
|||||
|
N |
Minimum |
Maximum |
Mean |
Std. Deviation |
Age |
106 |
24,0 |
75,0 |
54,340 |
11,7796 |
Valid N (listwise) |
106 |
|
|
|
|
#11 “The primary variable of interest” sounds like a dependent variable. However, I assume the authors used DAT as their independent variable. This needs to be revised.
- We analysed the diagnosis-to-ablation time (DAT), defined as the time in years between the initial diagnosis of AF and the first AF CA. In the primary analyses, DAT was treated as a continuous variable in years and reported as a median for both the group with recurrence and the group without recurrence. DAT was also considered as a categorical variable and patients were stratified into 3 categories: DAT<1 year, DAT<2 years and DAT<4 years
#12 “Other variables included ..” – It is unclear whether the listed variables were used as covariates in the confounding model. I suggest the authors perform a multivariable logistic regression analysis as the primary analysis and include the covariates listed.
- The covariates list is shown in Table 4, we used Cox multivariable regression and presented the diagnostics of the model
#13 The information in paragraph 2.4 seems a bit out of place. I suggest this information should be included in the study population.
- We modified the paragraph as you suggested;
Results
#14 The baseline characteristics displayed in Table 1 should be presented in the full study cohort (as it is done by the authors) and by the exposure groups.
#15 As already stated, the primary analysis needs to be revised. The authors are requested to perform a multivariable analysis on the association between diagnosis-to-ablation time and the recurrence of atrial fibrillation. Instead of Wilcoxon-Mann-Whitney and Kruskal-Wallis or Fisher tests, I prefer that the authors perform regression analyses and display the odds ratios.
#16 Please also revise Tables 2 and 3, respectively, both of which should show odds ratios.
- Our idea was (as seen in other studies with this model) to perform first the univariate analysis for all the variables (table 2 and 3) and then to do the Cox regression(table 4) as we were advised by the statistics team to do; the Fishers and WMW were done in a first step in order to better select which variables to include in the regression.
#17 For the categories of the diagnosis-to-ablation time, the authors are asked to provide odds ratios in the text.
- The multivariable analysis was performed as already stated (using Cox proportional hazards, in Table 4 the Exp (B) is shown; due to the relatively small number of events in a multivariable regresion model we were advised by the statistics crew to introduce only 5 covariates; but we modified the table 3 in order to show the odds ratio as you requested and if you wish we may delete Table 2 if you think it is not necessary.
Discussion
#18 I suggest that the authors restructure the discussion towards a more results-focused discussion. For example, at the beginning of the discussion, the authors should focus on stating their main findings.
- We restructured the discussion as advised;
#19 It is not entirely clear how the authors explain the results of their categorical DAT variable.
- We tried multiple cutoff values for DAT,and coded them „0” below the cutoff and „1” above the cutoff value: 1 year, 2 years and 4 years, meaning we created, (besides the numeric variable DAT (actual number of years from first diagnosis to ablation)), three more variables that were categorical, like this: „DAT_1year”: all the patients that had more than 1 year since first diagnosis to ablation were coded „1” for this variable and those that had less than 1 years were coded „0” etc, like in the table below:
Example:
Patient name |
DAT(numeric,in years) |
DAT_1year |
DAT_2years |
DAT_4years |
XXXX |
3 |
1 |
1 |
0 |
XZYX |
0.5 |
0 |
0 |
0 |
ZZZZ |
7 |
1 |
1 |
1 |
YYYY |
1.5 |
1 |
0 |
0 |
Limitations
#20 The limitations section should be expanded: What about data availability, generalizability, heterogeneity of study groups, residual confounding, and other potential biases? Can the authors please comment on this?
- The data are available on request; these are „real world” data from a medium size center in a country that is in confronted with severe finacial limitations, in which the patient’s access to CA for AF is limited, thus perhaps older patients (maximum age in the lot was 75 years old) and those with more severe comorbidities were not included.
Conclusions
#21 Based on the revised analysis, I suggest that the authors change the conclusion statement.
- We changed the conclusion statement
Reviewer 4 Report
Comments and Suggestions for Authors
Dear Authors,
Dear Authors,
our manuscript is well written, scientific language used is appropriate and does not present conceptual errors, but the originality is low. The concept of the early rhythm control strategy was already highlighted in studies like for example the EAST-AFNET. Moreover the role of DAT in predicting the failure of AF ablation was described in previous studies with lerger population than yours, setting the threshold of the upper limit at 1 or 3 years, depending on the single study. You set the time threshold at 2 years. I noted that the percentage of patients with persistent atrial fibrillation is doubled in the group of failure of AF ablation. It will be interesting to analize DAT in the two different group (paroxysmal vs persistent AF) in order too see if the role of the DAT is different in those two population.
However as you rightly pointed out in the limitations paragraph this is an observational retrospective study and the number of patients included is limited. Anyway we can consider DAT as an important parameter to consider in our clinical practice to better identify the right therapetuic strategy.
Author Response
Dear Reviewer 4,
Thank you so much for taking the time to analyse our work and for your valuable comments, we acknowledge your conclusions but perhaps it is worth mentioning the fact that some controversies still exist on the topic, if you consider, for example the paper published by Kalman et al [35] defined 'early' AF ablation as occurring within 1 month and 'delayed' ablation as taking place within 12 months.
Round 2
Reviewer 3 Report
Comments and Suggestions for Authors
I thank the authors for providing a revised version of the manuscript. I think the manuscript has substantially improved and is now acceptable for publication. Thank you.
Reviewer 4 Report
Comments and Suggestions for Authors
Dear Authors,
thanks for the improved version of your manuscript.
The baseline characteristics of the population are now clearly showed in the table 1 (no difference in the proportion of persistent AF in patients with DAT < 2 years vs > 2 years). Your work reinforce the necessity to perform AF ablation as early as possible. I agree that the best window of opportunity for catheter ablation in the course of the disease is matter of future research.